# Breastfeeding-Related Health Benefits in Children and Mothers: Vital Organs Perspective

**DOI:** 10.3390/medicina59091535

**Published:** 2023-08-25

**Authors:** Julio César Muro-Valdez, Alejandra Meza-Rios, Blanca Rosa Aguilar-Uscanga, Rocio Ivette Lopez-Roa, Eunice Medina-Díaz, Esmeralda Marisol Franco-Torres, Adelaida Sara Minia Zepeda-Morales

**Affiliations:** 1Laboratorio de Análisis Clínicos y Bacteriológicos (Vinculación), Departamento de Farmacobiología, CUCEI, Universidad de Guadalajara, Boulevard Marcelino García Barragán, No. 1421, Guadalajara 44430, Mexico; julio.muro@alumnos.udg.mx (J.C.M.-V.); alejandra.mezarios@academicos.udg.mx (A.M.-R.); 2Laboratorio de Microbiología Industrial, Departamento de Farmacobiología, CUCEI, Universidad de Guadalajara, Boulevard Marcelino García Barragán, No. 1421, Guadalajara 44430, Mexico; 3Laboratorio de Investigación y Desarrollo Farmacéutico, Departamento de Farmacobiología, CUCEI, Universidad de Guadalajara, Boulevard Marcelino García Barragán, No. 1421, Guadalajara 44430, Mexico; 4Instituto Transdisciplinar de Investigación y Servicios, CUCEI, Universidad de Guadalajara, Av. José Parres Arias 5, Rinconada de la Azalea, Industrial Belenes, Zapopan 45150, Mexico

**Keywords:** breast milk, lactation, breastfeeding, human health

## Abstract

Breast milk (BM) is a constantly changing fluid that represents the primary source of nutrition for newborns. It is widely recognized that breastfeeding provides benefits for both the child and the mother, including a lower risk of ovarian and breast cancer, type 2 diabetes mellitus, decreased blood pressure, and more. In infants, breastfeeding has been correlated with a lower risk of infectious diseases, obesity, lower blood pressure, and decreased incidence of respiratory infections, diabetes, and asthma. Various factors, such as the baby’s sex, the health status of the mother and child, the mother’s diet, and the mode of delivery, can affect the composition of breast milk. This review focuses on the biological impact of the nutrients in BM on the development and functionality of vital organs to promote the benefit of health.

## 1. Introduction

BM is widely recognized as the benchmark for nourishing newborns. The World Health Organization (WHO) and the United Nations Children’s Fund (UNICEF) advocate for exclusive breastfeeding during the initial 6 months of an infant’s life, with the exception of water, and introducing complementary foods up to the age of 2 years [1,2]. Other WHO recommendations for breastfeeding management include initiating breastfeeding within the first hour after birth and feeding the baby on demand [3]. However, according to the Infant Feeding Area Graphs Interpretation Guide published by UNICEF, exclusive breastfeeding rates were reported to be only 44 percent globally in 2021 [1]. BM is a biofluid known to contain a diverse array of macro- and micro-nutrients, including proteins, carbohydrates, lipids, vitamins, and minerals. Furthermore, it includes bioactive compounds such as hormones, growth factors, digestive enzymes, transporters, and antimicrobial agents, as well as maternal cells, including leukocytes and stem cells [4]. Based on its rich composition, BM is presumed to fulfill the nutritional requirements of infants, facilitating optimal development, growth, and cognitive function [5]. The composition of BM behaves dynamically, constantly changing and adjusting to the infant’s needs over time. Several factors influence BM composition, including the duration of lactation, maternal health conditions, genetic factors, and dietary choices, among others [6].

Research has demonstrated that the consumption of BM offers a multitude of health benefits. These include the modulation of the gut microbiota, protection against pathogenic bacteria in the gut, and potential protective effects against diseases such as diabetes and obesity [7,8,9]. Additionally, BM has been shown to possess antiviral properties [10,11]. This article is a scoping review that provides specific benefits of BM for the brain, lungs, liver, kidneys, and heart, as well as its effect on the intestinal microbiota and the health benefits to the breastfeeding mother, based on studies registered in public databases as well as in specialized engines. The evidence discussed here is based on clinical trials and comes from a search that goes back to ten years before the publication date except for information or classic documents necessary for the basis of the review.

## 2. Breastfeeding Effects on Brain Health and Development

The correct diet in early life plays a decisive role in ensuring the proper functional and structural development of the central nervous system [12]. Numerous studies have investigated the impact of BM on brain development, with a particular focus on the effects of 2′-fucosyllactose (2′-FL). Recently, a study demonstrated that sialic acid derived from sialyllactose present in the milk freely crosses the blood–brain barrier, with higher concentrations observed in gangliosides and glycoproteins within the gray matter [13]. In this context, Vázquez et al. evaluated the effect of 2′-FL on synaptic plasticity and cognitive function in an animal model using C67BL/6 mice and Sprague–Dawley rats. The results revealed that animals treated with 2′-FL exhibited improved scores in cognitive tests, along with increased expression levels of molecules associated with short-term memory [14]. In 2020, Berger and colleagues demonstrated through the Bayley scale (which evaluates cognitive, motor, and linguistic abilities at 24 months of life) that the frequency of lactation in the first month of life is a key factor in the cognitive development of infants, associated with higher exposure to 2′-FL [15], suggesting the importance of early exposure to 2′-FL, especially within the window of the first 6 months when the brain mass reaches half of its adult size [15,16]. Tarr et al. evaluated the properties of human milk oligosaccharides (HMOs) on brain tissue by administering 3′Sialyllactose and 6′Sialyllactose to C57BL/6 mice for two weeks. The HMOs protected the mice from the deleterious effects of stressor factors, regulated the gut microbiota, and maintained the normal number of neurons in the brain convolutions; these results showed the pivotal role of BM in the gut-brain axis [17].

A publication from 2015 investigated the impact of the duration of BM consumption, revealing that infants who were breastfed for a minimum of 12 months exhibited an average increase of 3.76 IQ points in adulthood compared to infants who were breastfed for only 1 month [18]. While it is widely acknowledged that BM consumption offers short-term benefits, it is important to emphasize that breastfeeding can also yield long-term advantages for the infant.

In another study using magnetic resonance imaging, brain white matter volume was compared in infants who were breastfed for a minimum of 3 months, finding that infants who were breastfed had greater white matter development in brain regions linked to language, emotion, and cognition [19]. These results are consistent with the results obtained in a study where BM consumption was associated with increased white matter and greater cognitive development [20].

The phospholipids present in BM have also demonstrated beneficial effects on brain development. The precursors of choline sphingomyelin and phosphatidylcholine are a factor in normal memory and cognitive function, as well as brain and neurological development [21]. Comparative studies between BM and formula milk have shown that BM significantly increases the levels of sphingomyelin and choline in the brain [22]. Several components of BM, such as cholesterol, long-chain polyunsaturated fatty acids, and docosahexaenoic acid, among others, can provide benefits to the less-developed brains of preterm infants and enhance neurodevelopment [23]. In this context, Zhang et al. conducted a study to evaluate the effect of breastfeeding on early brain development in premature babies. Their observations revealed that BM increased regional gray matter volume in comparison to formula-fed babies, specifically in various brain regions, including the right temporal lobe, left caudate nucleus, and bilateral frontal lobe. Furthermore, BM was found to enhance brain activation in the superior temporal gyrus [24].

In a study aimed at investigating the target sites of cells present in BM, a BALB/c mice model was utilized, and fluorescently labeled cells were employed along with real-time PCR and immunohistochemistry techniques. Stem cells were found in the blood and brain, which were derived from BM; furthermore, these cells demonstrated the ability to differentiate into neuronal and glial cells. Based on these results, it has been postulated that breast milk stem cells might play a role in brain maturation and provide support to host cells through the secretion of growth factors [25].

One of the functions related to brain development is speech ability. Results of a meta-analysis conducted by Abida et al. show the effect of breastfeeding on speech development in children. The results of the analysis of cohort studies indicate that breastfeeding increases language development by 1.19 times compared to children who did not receive this type of feeding; the effect observed in cross-sectional studies was 1.54 times greater language development in breastfed children compared to those who did not [26]. The results are consistent with what was published by Rosidi and collaborators, who suggest that exclusively breastfed children acquire sufficient energy to favor this improvement in speech [27].

Recently, interventional clinical trials have used BM as a nutritional intervention for some specific conditions to enhance newborn brain development. Some of these clinical trials are summarized in Table 1 [14 March 2023 from: https://clinicaltrials.gov/].

## 3. Breastfeeding Benefits Kidney Performance

The current information on this subject is limited; however, the existing studies in this area are summarized below. Boutrid et al. conducted a systemic review to explore the association between breastfeeding and renal prognosis. The review revealed several notable findings. (a) New thresholds for calciuria were determined according to different child feeding, (b) lactation during the first 4 months of life was correlated with a lower estimated glomerular filtration rate (eGFR) and increased combined kidney volume (*p* < 0.05) (based on the analysis of over 5000 children) [28], (c) higher protein content in formula milk led to increased kidney size (42.6 cm^3^) in 6-month-old infants compared to infants who were breastfed and had a normal kidney size (19.1 cm^3^); however, the long-term effect of consuming high-protein formulas needs to be studied [28,29]. This suggests a possible negative impact of excessive protein consumption in the early stages of infancy. The author concluded that BM may have favorable renal effects during infancy [28]. There is a direct association between the amount of protein consumed and the growth and function of the kidneys, where the amount of protein consumed in the first months of life may promote different patterns of kidney tissue growth and, thus, affect the long-term function of these organs [29]. In this context, Miliku et al. explored the associations between breastfeeding duration and kidney outcomes during school age; the study included a total of 5043 children from the Netherlands and found that breastfed children versus never breastfed had smaller kidneys and presented lower eGFRs. In addition, shorter durations of breastfeeding were associated with smaller kidney volumes and a lower risk of microalbuminuria. Based on their findings, the authors concluded that breastfeeding is associated with subclinical changes in kidney outcomes during childhood [30].

The development of kidney stones has been associated with the type of diet, the type of fluids consumed, and micronutrients such as vitamin D and calcium. Urolithiasis in children has also been related to hereditary genetic factors and metabolic disorders [31,32]. Shajari et al. investigated the correlation between the type of milk consumed and the risk of developing kidney stones. The study involved 30 children below the age of two who had been diagnosed with kidney stones and 125 children without stones as a control group. The results showed a significant difference in the duration of breastfeeding between children with kidney stones and those without, concluding that breastfeeding may serve as a nutritional factor to prevent and protect against the development of kidney stones in children [31]. Recently Bozkurt et al. evaluated the potential impact of breastfeeding duration on the clinical progression and therapy of kidney stones detected in infancy. The study included 48 children with renal stones, and the duration of lactation was evaluated, taking into account the characteristics of the stones at the time of the diagnosis. The findings demonstrated that the duration of breastfeeding was longer in patients who did not experience disease progression and in infants with a smaller size and/or fewer stones. Furthermore, according to this study, children who exclusively breastfed for the first six months of life required less treatment and exhibited lower rates of growth retardation [32].

## 4. Breastfeeding and the Lung Function

The influence of exclusive breastfeeding on lung function in a later stage of life is still a topic of controversy. Several studies have explored the potential correlations between breastfeeding and lung function in children who were exclusively breastfed for an extended duration [33]. Furthermore, investigations into the lung microbiome have revealed insights into the relationship between respiratory diseases and lung microbiota, highlighting the favorable effects of prolonged exclusive breastfeeding on the function of smaller airways throughout childhood. Observational studies have also suggested that breastfeeding mitigates the lung consequences of respiratory infections, leading to improved lung function during school age, particularly among children with atopic backgrounds. Moreover, research has indicated that breastfeeding can help protect lung function in individuals who are exposed to high levels of air pollution [34], including passive smoking [35]. These findings suggest that breastfeeding might mitigate the negative impacts of environmental factors on the developing lungs [36].

The protective influences of breastfeeding on lung function may be attributed to a decrease in respiratory infections [37] and the association with increased height in breastfed children [38]. Moreover, the positive impact of longer breastfeeding duration on lung capacity, as measured at 10 years old, persists until 18 years of age [37]. This is of particular significance considering that asthma, the highest prevalent chronic non-communicable childhood disease, affects approximately 14% of children worldwide [39], leading to frequent emergency room visits and school absenteeism, often resulting in lifelong impairments in lung health [40]. Several investigations have demonstrated that certain HMOs have the ability to directly modulate the immune response by mitigating viral pathogens and modulating the behavior of immune cells within the host [41]. These findings highlight the potential immunomodulatory properties of HMOs concerning respiratory health.

Furthermore, there is an additional potential mechanism through which HMOs provide protection against asthma. Around 1% of HMOs are absorbed into the bloodstream and eventually reach target organs, including the lungs [42]. HMOs can influence the turnover of airway epithelial cells, the formation of glycocalyx mucus, and interact with immune cells and pathogens, thereby offering protection against asthma [43]. Many factors could contribute to the effect of breastfeeding on lung function outcomes. Breastfeeding plays a role in preventing respiratory tract infections, which, in turn, promotes optimal lung growth. Additionally, the reduced accumulation of body fat in breastfed infants may contribute to enhanced lung performance [38].

Several studies have reported conflicting results. In a research study conducted on healthy children with minimal risk of asthma, the duration of breastfeeding was not found to be associated with lung function outcomes at 6 years of age [38,44]. These divergent findings may be attributed to variations in the size of the sample, the focus of determinants, participant age, confounding factors, and study methodologies. On the other hand, the literature also mentions a non-immunological effect of BM. For instance, Castellote et al. highlighted that colostrum contains abundant growth factors such as TGF-β, and the concentration of these molecules gradually decreases in BM during the initial months of life [45]. These growth factors have the potential to promote positive lung development by enhancing elastin activity in fibroblasts [37,46].

## 5. Breastfeeding Effects on the Liver Tissue and Hepatic Functions

The liver plays a vital role as the primary organ responsible for biochemical metabolism within the human body (energy metabolism and the maintenance of metabolic homeostasis). It metabolizes various compounds, both beneficial and harmful, as well as endogenous and exogenous molecules. Compounds absorbed by the intestine, including drugs and nutrients, first pass across the liver, resulting in the formation of smaller products and the regulation of their blood levels [47,48]. The shift from being a fetus to becoming a newborn represents the most intricate adaptation that takes place in the human journey [49]. The liver of a newborn must swiftly adjust and develop to meet the demands presented by life outside the uterus. This maturation process involves various molecular, cellular, and environmental factors, as identified previously by Chen et al. [50]. The consumption of milk and, later, the introduction of solid foods provides a greater abundance of nutrients and requires the neonate to take charge of its metabolic requirements, consequently influencing postnatal hepatic metabolism [51,52,53]. A pioneering study focused on breast milk’s impact on liver development was conducted by Kohno et al. They researched the influence of BM on DNA synthesis in rat neonatal hepatocytes to assess its physiological role in liver growth. The results demonstrated that BM increased DNA synthesis; also, the BM presented a mitogenic activity in vitro, suggesting its potential importance for the growth and development of the infant liver [54].

Carvalho et al. studied the impact of postnatal diet (BM versus formula milk) on liver mitochondrial bioenergetics phenotypes in male piglets. The researchers explored how this “programming” of physiological systems could have potential metabolic consequences in childhood and adulthood. The piglets fed with milk formula exhibited higher ADP-linked respirations, suggesting increased ATP turnover in the liver, which in turn drove an elevation in oxygen consumption [55]. In this context, Pena–Leon et al. investigated the impact of breastfeeding on the infant’s reprogramming of energy balance during childhood and adulthood. They utilized a rat model where prolonged breastfeeding or suckling was followed by feeding the rats either a chow diet or a high-fat diet until adulthood. The results revealed that extended breastfeeding could mitigate the consequences of diet-induced obesity by serving as a persistent physiological stimulus for thermogenesis in brown adipose tissue. The activation of this program stimulated energy expenditure, leading to reductions in adiposity, dyslipidemia, and weight gain. The proposed mechanism that underlies this protective effect involves the liver, specifically by promoting elevated expression and secretion of hepatic FGF21. This molecule is able to access the central nervous system, where it regulates systemic effects and reprograms hypothalamic circuitries, rendering the descendants additionally resistant to diet-induced obesity later in life [56]. On the other hand, a comparison of serum biochemistry between breastfed children and formula-fed children revealed significant differences in molecules directly related to liver metabolism. The breastfed group exhibited higher levels of cholesterol, triglycerides, alanine aminotransferase, aspartate aminotransferase, gamma-glutamyl transferase, total bilirubin, and direct bilirubin compared to the formula-fed infants. These findings indicate that different sources of nutrition may cause distinct metabolic responses [57].

Conversely, Gart et al. investigated the hepatoprotective properties of HMO (2′-FL) in an animal model of non-alcoholic liver disease (NAFLD). The results showed that supplementation with 2′-FL acted as a hepatoprotective molecule in the liver, effectively suppressing the accumulation of lipids or microvesicular steatosis, which is associated with obesity. The treatment also improved lipid handling, demonstrated by functional transcriptome analysis. In addition, 2′-FL was found to reduce specific bioactive lipids in the liver, such as diacylglycerol, which is known to affect insulin signaling and play a role in ER stress. Diacylglycerol is also a significant factor in the dysmetabolic state that leads to insulin resistance [58].

## 6. Breastfeeding Effects on Cardiovascular Tissue

The heart is one of the vital organs responsible for blood distribution across the body. Currently, it is well known that breastfeeding has protective effects against childhood obesity [59,60] and cardiovascular disease risk factors [61]. Some studies have demonstrated that BM led to a small reduction in adult blood pressure levels, reduced total cholesterol and LDL cholesterol levels, and modified adult body mass index (BMI) [62]. BM has antibodies, stem cells, hormones, HMOs, growth factors, and enzymes which have the potential to enhance cardiovascular tissue development during the neonatal and infant stages [63].

Based on the BM’s composition, the potential mechanisms that could reduce long-term cardiovascular risk focus on molecules, including vascular endothelial growth factor (VEGF) and adiponectin [63]. Animal model systems have demonstrated that VEGF participates in many different aspects of cardiovascular development, including endothelial cell differentiation, survival, and migration, including heart formation and hematopoiesis [64]. Adiponectin is a major component of BM and has demonstrated direct effects on the vasculature and the heart. It also helps mitigate oxidative stress in endothelial cells, promotes the mobilization and function of endothelial progenitor cells, and inhibits apoptosis, oxidative stress, and inflammation in cardiomyocytes [63]. In addition, stem cells found in BM have demonstrated the ability to differentiate into cardiomyocytes and integrate into different tissues, contributing to normal development [63,65]. As described, HMOs play a role in maintaining hemodynamics and supporting the proper development of heart tissue and the vascular system [63,66].

Furthermore, Umer et al. evaluated the correlation between childhood cardiovascular disease (CVD) risk factors and breastfeeding in 10,457 children in fifth grade with a mean age of 11.0 ± 0.5 years. The outcome variables in the study included blood pressure and lipid profile. The results showed that children who were fed with BM had significantly smaller mean triglyceride amounts when compared to individuals who were not breastfed. Biological samples and participant data were collected at an average age of 11 years [67].

In contrast, the long-term cardiovascular consequences of premature birth include reduced bi-ventricular volume, reduced systolic and diastolic function, disproportionate growth of muscle mass, higher probability of heart disease, and lower exercise tolerance, among others. BM might exhibit protective effects by reducing these pathophysiological alterations and preventing the onset of CVD in adulthood [63]. Exclusively breast milk-fed preterm infants present a reduction in cardiac structural parameters as adults compared with those who received formula milk, indicating a protective effect of BM on long-term cardiac phenotype [68].

Finally, interventional clinical trials that are recruiting, enrolling by invitation, active, unknown, not recruiting, or completed using breast milk as a treatment for cardiovascular diseases are summarized in Table 2 [14 March 2023 from: https://clinicaltrials.gov/].

## 7. Comparison between Breastfeeding and Non-Breastfeeding in Gut Microbiota Health and Outcomes

The term microbiota refers to the collection of microorganisms that reside in a particular environment and share a common habitat [69]. The human microbiota is formed by different types of microorganisms, including bacteria, fungi, viruses, and other unicellular organisms living in diverse anatomical zones [69,70]. About 80% of the microbiota found in healthy adults consists of Bacteroidetes and Firmicutes phyla, together with others like Proteobacteria, Fusobacteria, Actinobacteria, and Verrucomicrobia [69,71]. The factors that can affect microbiota composition in childhood and adult life include the method of delivery (cesarean section or vaginal), BM or formula milk feeding, diet, exposure to drugs/antibiotics, presence of siblings in the household, presence of furry pets in the household, geographical location, and others [72].

Victora et al. conducted meta-analyses based on 28 systematic reviews to investigate the connections between breastfeeding and outcomes in children or mothers. The results indicated that breastfeeding provides protection against childhood infections, reduces the risk of dental malocclusions, enhances intelligence, lowers the risk of overweight and diabetes, and, overall, reduces infectious morbidity and mortality [73]. It is well known that healthy gastrointestinal flora is responsible for the complete health of the host [74]. In addition, breastfeeding has important effects on microbiota composition. Infants fed with BM have a dynamic gut microbiome and present lower incidences of certain diseases [75]. BM serves as a rich source of bacterial species, harbors its microbiome, and plays a crucial role in introducing beneficial bacteria to the infant’s gastrointestinal tract after birth. It is throughout the first thousand days of life that a child’s gut microbiota is established, and any disturbances or altered colonization during this neonatal period can potentially impact health outcomes later in life [75,76]. Some of the microbiota mechanisms of action in the gut are the production of antimicrobial molecules, enhancing intestinal mucin production, and preventing the adhesion of pathogenic bacteria [75,77].

On the first day of life, the gut microbiota of newborns delivered vaginally closely resembles that of the maternal vagina and intestinal tract, in contrast to those born via cesarean section; their microbiota is like maternal skin. The gut microbiota at birth is characterized by its low variety. By 1–2 years of age, the microbiota is more complex than the gut microbiota of adults; therefore, the initial year of life plays a crucial role in the establishment of microbiota, with BM being the primary influencing factor in terms of composition and its long-term impact on health [72]. BM is a bioactive substance that acts as a prebiotic and probiotic due to its oligosaccharides and bacteria that are not yet mimicked in formula milk [5]. Lactation influences health-promoting microorganisms mediated by factors like antibacterial peptides, components of the innate immune system, and polymeric IgA. BM also improves the mucosal defenses and barrier integrity of the intestinal tissue [72,78,79]. Ma et al. compared gut microbiota from 91 healthy children that were totally fed with BM or different types of formula milk for more than four months. The findings indicated that breastfed groups exhibited lower alpha diversity compared to formula-fed groups at 40 days of age, but this diversity increased significantly by the time the infants reached 6 months of age. The predominant genera were *Bifidobacterium* and *Enterobacteriaceae* in babies with 40 days of life, *Bacteroides* and *Bifidobacterium* were higher in the breastfed group, and *Streptococcus*, *Lachnospiraceae*, *Veillonella*, *Clostridioides,* and *Enterococcus* were lower compared to the formula-fed group [72]. *Bifidobacteria* has a major role in the development of the immune system, preventing infections in children. In contrast, *Bifidobacterium* has multiple health benefits, including vitamin production, modulation of the immune system, reduction in the prevalence of atopic dermatitis and rotavirus infections, and lower lactose intolerance in children and adults [72,80]. In addition, *Bifidobacteria* abundance indicates a better immune response to vaccination and is correlated with a reduced risk of obesity and allergic diseases [72,81,82]. *Bacteroides* is another bacteria that, in the earlier neonatal phase, participates in the growth of the mucosal immune system. These mechanisms may confer lifelong protection against health disorders in the human body. This beneficial bacteria is also linked with increased diversity and faster maturation of the gut tissue [72]. Among the bacteria with lower abundances in children fed with BM is *Streptococcus* sp. Some studies have shown that higher levels of this genus are seen in patients with type 1 diabetes [72,83].

The gut microbiota and brain develop together during the first year of life. Gut microbiota is recognized as a modulator of behavior, including cognition and social skills [84]. In addition, several researchers have demonstrated that the gut microbiota also impacts neurodevelopment in the first year of age, reducing the risk of acquiring neurodevelopmental and neuropsychiatric diseases [5,85]. Research has indicated that an increased abundance of Bacteroides and a decreased abundance of *Shigella*/*Escherichia* and Bifidobacterium are negatively associated with fine motor skills in children. Similarly, elevated levels of certain taxa from *Lachnospiraceae* and *Clostridiales* and reduced levels of Bacteroides are negatively correlated with communication and social skills in children at the age of 3 [86]. Carlson et al. published results from a pilot study in infants exclusively breastfeeding until 1 month of life whose microbiota composition showed an association with non-social fear behavior [87].

## 8. Lactation Benefits in Mother’s Health

Longer lactation has been correlated with long-term health; mothers can experience various benefits, including a reduced risk of heart disease (high blood pressure), some types of cancer (endometrial, ovarian, and breast cancer), metabolic syndrome, NAFLD, hypercholesterolemia, and type 2 diabetes mellitus [73,88,89,90]. The short-term benefits of lactation in women’s health include reductions in infectious symptoms, stress responsivity, blood pressure, weight loss, better positive moods, and fertility control; all of them related to parasympathetic activation, endocrine factors, and oxytocinergic mechanisms present in the breastfeeding period [89].

Studies showed a correlation between breastfeeding and a reduced relative risk of breast cancer in parous women. Specifically, for every 12 months of breastfeeding, the relative risk of breast cancer decreases by 4.3%. Additionally, parous women who have breastfed at any point in their lives have a 14% lower risk of developing breast cancer compared to those who have never breastfed. The protective effect of breastfeeding is more pronounced in women who have breastfed for a cumulative duration of 12 months or longer, with a 28% lower risk of breast cancer [90]. Based on the meta-analysis conducted by Victora et al. published in 2016, which included 50,000 patients with cancer, for every additional 12 months of breastfeeding over a woman’s lifetime, there was a 4.3% decrease in the occurrence of invasive breast cancer [73]. The authors estimated the potential effect of lactation on breast cancer mortality. In their estimation, the current global rates of breastfeeding prevent 19 breast cancer deaths annually, compared to a scenario where no women breastfeed. To summarize, the current global rates of breastfeeding prevent approximately 20,000 deaths from breast cancer each year [73]. However, Lambertini et al. reported that breastfeeding was significantly associated with lower odds of developing luminal and triple-negative breast cancer subtypes but with no difference in the human epidermal growth factor receptor (2HER2) breast cancer subtype [91].

Alternatively, Victora et al.’s meta-analysis included 41 studies for ovarian cancer. They reported a 30% decrease correlated with prolonged periods of breastfeeding [73]. In 2020, Babic et al. published the results of a pooled analysis of 13 case-control studies involving parous women with ovarian cancer and controls with the objective of determining the association between lactation and this type of cancer. The analysis included a total of 9973 women with ovarian cancer and 13,843 controls; the results indicated that women who had ever breastfed had a decreased risk of developing all types of invasive ovarian cancers, with a particularly notable reduction in the risk of endometrioid and high-grade serous subtypes (which are the most lethal types of ovarian cancer). Women who breastfed for 1 to 3 months were related with 18% lower risk, and lactation for ≥12 months was correlated with a 34% lower risk. This reduction in risk persists for decades. The author concluded that the findings indicate that breastfeeding could be a factor that can be modified and has the potential to reduce the risk of ovarian cancer [92].

For diabetes mellitus type 2, six cohort studies showed an odds ratio of 0.68 [73,93]. In addition, an association analysis between overweight and breastfeeding that included 740,000 British women with long-term follow-up revealed that mean BMI was 1% less for every 6 months of lactation [73,94]. Other meta-analyses showed that the longest period of lactation was correlated with a 32% decreased relative risk of type 2 diabetes versus the shortest period of lactation [95]. On the other hand, Pinho–Gomes et al. conducted a systemic review and meta-analysis research to evaluate the correlation between lactation and maternal risk of type 2 diabetes. Longer-term studies have revealed a consistent and gradual protective association between lactation and the risk of type 2 diabetes. The risk reduction appears to be more pronounced in mothers with gestational diabetes compared to those without. Breastfeeding versus never breastfeeding was correlated with a 27% lower risk of this pathology, and each additional month of breastfeeding was found to be associated with a 1% decrease in the risk of developing type 2 diabetes [96]. In 2018, Gunderson et al. published results from the 30-year Coronary Artery Risk Development study, aiming to address the key question, “Is there a biochemical evidence basis supporting the protective association between lactation duration and progression to gestational diabetes?”. The study enrolled 1238 women, and the association analysis revealed a robust and graded inverse relationship between the incidence of diabetes and lactation duration. The findings indicated an increased risk of developing diabetes among women who did not engage in lactation compared to those who lactated for 12 months or more [97].

Tschiderer et al. published a systemic review and meta-analysis of eight studies and more than 1 million parous women, and they found that lactation prevents future stroke, coronary heart disease, and fatal CVD. The authors also found a progressive reduction in cardiovascular risk with breastfeeding durations of up to 12 months throughout life [98]. In accordance, researchers have shown that breastfeeding was correlated with lower odds of hypertension and a lifetime lactation duration of >6 months was significantly associated with this outcome [95]. Furthermore, non-breastfeeding women have a higher risk of vascular changes associated with future cardiovascular diseases like calcified atherosclerotic plaques and higher carotid adventitial diameter. Evidence showed that mothers who never breastfed had larger adventitial diameters and carotid artery lumen versus mothers who breastfed. These characteristics are indicative of poorer cardiovascular health status [95,99]. A study with middle-aged and elderly women with a lifetime lactation duration longer than 12 months presented a reduced risk of incident myocardial infarction versus parous women who never breastfed. In addition, breastfeeding appeared to reduce mortality from ischemic heart disease. Women aged over 65 years who have never breastfed have an approximately threefold higher risk of CVD mortality over a 15-year period compared to women who have breastfed for more than 24 months in their lifetime [95,100,101].

In contrast, a study conducted by Veeral et al. demonstrated that breastfeeding for a longer duration, specifically more than 6 months, is associated with a reduced risk of non-alcoholic fatty liver disease (NAFLD) in mid-life. The study followed 844 women over a period of 25 years, with 32% reporting a breastfeeding duration of 0 to 1 month, 25% breastfeeding for 1 to 6 months, and 43% breastfeeding for more than 6 months. The results revealed an inverse relationship between longer lactation and the development of NAFLD in mid-life, particularly breastfeeding for more than 6 months. This finding suggests that breastfeeding duration may serve as a modifiable risk factor for NAFLD. The authors propose that the mechanism underlying this association could be attributed to the long-lasting effects on body fat distribution, insulin sensitivity, and circulating lipid levels [88].

## 9. Conclusions

Breastfeeding has beneficial effects on infant and maternal health, as breast milk is the main source of newborn nutrition. The current evidence obtained from observational studies points out the strong role between breastfeeding and improvement in long-term vital organ outcomes. Breastfeeding is the best choice of diet for a newborn. Promotion of breastfeeding during the perinatal period may represent an exceptional chance to reduce the risk prevalence of health problems until adulthood. However, it is necessary to clarify the metabolic pathways that lead to the benefit of breast milk consumption from birth.

## Figures and Tables

**Table 1 medicina-59-01535-t001:** Clinical trials using breast milk to enhance newborn brain development.

Status	Title	Intervention	Country	Identifier
Recruiting	Targeting breast milk fortification to improve preterm infant growth and brain development	Individually targeted fortification	USA	NCT03977259
Completed	Supplemental choline and brain development in humans	Phosphatidylcholine Corn oil placebo	USA	NCT00678925
Completed	OptiMoM kindergarten study	Pasteurized donor human BM Preterm formula	Canada	NCT02759809
Active, not recruiting	Renoir: a randomized, double-blind, controlled trial to evaluate the effects of a new breast milk fortifier on growth and tolerance in preterm infants.	Test product Control product	France, Germany, The Netherlands, and the United Kingdom	NCT03315221

BM, breast milk.

**Table 2 medicina-59-01535-t002:** Clinical trials using breast milk as a strategy for cardiovascular diseases.

Status	Title	Intervention	Country	Identifier
Completed	Intestinal function in neonates with complex congenital heart disease	NPO Fresh BM	USA	NCT01475357
Active, not recruiting	Exclusive breast milk feeding in infants with single ventricular physiology	BM Derived Fortifier Human/Bovine Milk	USA	NCT02860702
Active, not recruiting	Internasal breast milk for intraventricular hemorrhage	BM	Canada	NCT04225286
Unknown	Effect of breastfeeding optimization on early vascular development	Breastfeeding optimization Usual care	Indonesia	NCT01566812
Active, not recruiting	Lower protein intake and long-term risk of obesity and cardiovascular disease	Modified infant formula Standard infant formula	London	NCT03456934

NPO, nothing by mouth; BM, breast milk.

## Data Availability

Not applicable.

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
