# Peer review of "Breastfeeding-Related Health Benefits in Children and Mothers: Vital Organs Perspective"

_medicina, 2023, doi:10.3390/medicina59091535_

Round 1

Reviewer 1 Report

Dear Editor
My overall evaluation of the manuscript is positive. There are a number of minor revisions, formal and scientific aspects that should be addressed.

1.      It is necessary to talk about the WHO guidelines on breastfeeding. At the same time, talk about the strategies that can be used for the implementation of WHO guidelines in the article.

2.      In addition, it is necessary to discuss the effect of breastfeeding on the epigenetic process in the expression of genes in the article.

3.       About the process of breastfeeding and its effect on the speech development in children should also be added to the article.

Author Response

Reviewer #1:

  1. It is necessary to talk about the WHO guidelines on breastfeeding. At the same time, talk about the strategies that can be used for the implementation of WHO guidelines in the article.

Other WHO recommendations on breastfeeding have been included in the brief (line 36-39).

  1. In addition, it is necessary to discuss the effect of breastfeeding on the epigenetic process in the expression of genes in the article.

Thanks for the comment, although our type of feeding affects epigenetic processes, the objective of the article is the effect of breast milk on the functioning of vital organs without going into fine molecular mechanisms, we believe that the epigenetic process is of great interest, but it is beyond the objective of the article.

  1. About the process of breastfeeding and its effect on the speech development in children should also be added to the article.

Thank you for your comment, in the section on brain development, in lines 115-123 we have added information on the effect of breastfeeding on brain speech capacity.

Reviewer 2 Report

Medicina

Type of manuscript: Review
Title: Breastfeeding-related health benefits in children’s and mothers:
vital organs perspective

 To the Editors and authors,

 Thank you for the invitation to review this manuscript. It was a pleasant reading. This is a very ambitious review, encompassing multiple organs functions, both in children and in their mothers. I will report my commentaries below.

General:

While I was reading the text, I missed comments on the reports of the studies cited, such as possibilities of confounding. I don’t know if there was a limitation of words, but this review deserved a more thorough approach. I recognize that they have an extensive list of references; it wouldn’t be possible to do it with all of them. But some comments could be made.

The authors could finish each topic with a summary of the actual knowledge on the subject. There are many advantages in the human milk composition; but, for example, regarding child’s development, environment, parents schooling, experiences, all this influence the final performance. One gets the impression that it is only necessary to give a specific nutrient encountered in higher proportion in human milk to improve development. The same is valid for all the others organs functions: kidneys, heart, intestine... We know how important is to offer small amounts of milk by mouth after birth to stimulate the growth of the intestine in preterm – and if we can use human milk, colostrum, this is much better; it even has imune properties, and favours a better microbioma.

Abstract and Introduction: It is clear.

Line 112 – 114: “Currently some interventional clinical trials used the BM as a treatment for some specific conditions to enhance newborn brain development.” Is it treatment really? Or they were studies of nutritional intervention?

Line 115 – 118: Table 1. Title: “Clinical trials under recruiting; enrolling by invitation; active, unknown, not recruiting; or completed status using breast milk to enhance newborn brain development [march 14th from: https://clinicaltrials.gov/].” Is this the title of this table? It looks like a legend. I missed a title describing the content of the Table.

Regarding the trials shown, I missed the period date.

Line 123 – 130, reference 25 (Boutrid et al): Were the diferences found statistically significant?

Line 126 - 127: “higher protein content in formula milk led to increased kidney size in 6-month-old infants compared to infants who were breastfed and had a normal kidney size.” What is “a normal size”? So, those infants fed formula had an abnormal kidney size? What about renal function? It needs clarification. And then, in the next paragraph, they present another study, from Miliku et al, reporting similar results. The presentation of both studies, and the authors conclusion, needs comments. And this study of Miliku et al should be described in the same paragraph of the study of Boutrid et al, as they are discussing the same subject.

Line 137: The authors begin the description of the study by Shajari et al, on kidney stones. They should separate it, starting a new paragraph.

Line 137 – 149 – Here the authors are discussing the relation of kidney stones and breastfeeding. I missed an explanation on the causal relationship, by both studies: Shajari et al and Bozkurt et al.

Line 149 – 151: “Furthermore, children who exclusively breastfed for the first six months of life required less treatment and exhibited lower rates of growth retardation [28]”. The authors should point that this refers to Boskurt et al study,

Line 153 – 158: This paragraph and part of the next one do not have any link with the topic under discussion: lung function. Probably they should be included in the introduction.

Line 159: “However, the influence of exclusive breastfeeding on lung function in later stage of life still a topic of controversy.” This topic should start with this frase, without the “However”.

Lines 282 – 284: Umer et al study – were the findings described significant? Which are the ages of the childrens evaluated? Was the trigliceride measured while the children were under breastfeeding? Or afterward, in older ages? This needs clarification.

Lines 296 – 299: Table 2. Same comments as for Table 1.

Writing:

This manuscript needs a revision of the english writing.   

It needs revision, mainly punctuation

Author Response

Reviewer #2:

  1. Line 112 – 114: “Currently some interventional clinical trials used the BM as a treatment for some specific conditions to enhance newborn brain development.” Is it treatment really? Or they were studies of nutritional intervention?

Thanks for the comment, the term has been corrected to nutritional intervention (line 124-125).

  1. Line 115 – 118: Table 1. Title: “Clinical trials under recruiting; enrolling by invitation; active, unknown, not recruiting; or completed status using breast milk to enhance newborn brain development [march 14th from: https://clinicaltrials.gov/].” Is this the title of this table? It looks like a legend. I missed a title describing the content of the Table.

Regarding the trials shown, I missed the period date.

The title has been shortened to Clinical trials using breast milk to enhance newborn brain development (line 128-129).

  1. Line 123 – 130, reference 25 (Boutrid et al): Were the diferences found statistically significant?

The p value (p<0.05) was included as published in the reference.

  1. Line 126 - 127: “higher protein content in formula milk led to increased kidney size in 6-month-old infants compared to infants who were breastfed and had a normal kidney size.” What is “a normal size”? So, those infants fed formula had an abnormal kidney size? What about renal function? It needs clarification. And then, in the next paragraph, they present another study, from Miliku et al, reporting similar results. The presentation of both studies, and the authors conclusion, needs comments. And this study of Miliku et al should be described in the same paragraph of the study of Boutrid et al, as they are discussing the same subject.

The consumption of formulas with high protein content favors the increase of kidney volume in infants, compared to breastfed infants and even with formulas with low content of this macronutrient. The long-term effect remains to be investigated. The kidney volume values for each study group have been added to the brief (line 140-148). 

The information concerning kidney size has been unified in a single paragraph.

  1. Line 137: The authors begin the description of the study by Shajari et al, on kidney stones. They should separate it, starting a new paragraph.

Thank you for your comment, the suggestion to separate the information has been made.

  1. Line 137 – 149 – Here the authors are discussing the relation of kidney stones and breastfeeding. I missed an explanation on the causal relationship, by both studies: Shajari et al and Bozkurt et al.

We have included the following information from the cited studies on the effect of lactation on the development of urinary tissue stones. "The development of kidney stones has been associated with the type of diet, the type of fluids consumed, and micronutrients such as vitamin D and calcium" (line 164-166).

  1. Line 149 – 151: “Furthermore, children who exclusively breastfed for the first six months of life required less treatment and exhibited lower rates of growth retardation [28]”. The authors should point that this refers to Boskurt et al study.

It has been clarified that the information is about what is described by Boskurt et al.

  1. Line 153 – 158: This paragraph and part of the next one do not have any link with the topic under discussion: lung function. Probably they should be included in the introduction.

The suggested information has been removed.

  1. Line 159: “However, the influence of exclusive breastfeeding on lung function in later stage of life still a topic of controversy.” This topic should start with this frase, without the “However”.

The beginning of this subtopic has been reconfigured according to your observation.

  1. Lines 282 – 284: Umer et al study – were the findings described significant? Which are the ages of the childrens evaluated? Was the trigliceride measured while the children were under breastfeeding? Or afterward, in older ages? This needs clarification.

Thank you for your observation. The average age of the participants was 11 years, and the data were collected at this age. The differences observed in the established parameter were significant. These comments have been included in the article (line 320-325).

  1. Lines 296 – 299: Table 2. Same comments as for Table 1.

The title has been changed to “Clinical trials using breast milk as a strategy for cardiovascular diseases”.

  1. Writing: This manuscript needs a revision of the english writing.  

English revision has been performed.
